# Design and Validation of Nanofibers Made of Self-Assembled Peptides to Become Multifunctional Stimuli-Sensitive Nanovectors of Anticancer Drug Doxorubicin

**DOI:** 10.3390/pharmaceutics14081544

**Published:** 2022-07-25

**Authors:** Valentina Del Genio, Annarita Falanga, Emilie Allard-Vannier, Katel Hervé-Aubert, Marilisa Leone, Rosa Bellavita, Rustem Uzbekov, Igor Chourpa, Stefania Galdiero

**Affiliations:** 1Department of Pharmacy, School of Medicine, University of Naples ‘Federico II’, Via Domenico Montesano 49, 80131 Naples, Italy; valentina.delgenio@unina.it (V.D.G.); rosa.bellavita@unina.it (R.B.); 2EA 6295 Nanomédicaments et Nanosondes, Faculté de Pharmacie, Université de Tours, 31 Avenue Monge, 37200 Tours, France; emilie.allard@univ-tours.fr (E.A.-V.); katel.herve@univ-tours.fr (K.H.-A.); 3Department of Agricultural Science, University of Naples ‘Federico II’, Via Università 100, 80055 Portici, Italy; annarita.falanga@unina.it; 4Institute of Biostructures and Bioimaging-CNR, 80145 Naples, Italy; marilisa.leone@cnr.it; 5Plateforme Scientifique et Technique “Analyse des Systèmes Biologiques” (PST ASB), UFR de Médecine, 37032 Tours, France; rustem.uzbekov@univ-tours.fr; 6Faculty of Bioengineering and Bioinformatics, Moscow State University, 119992 Moscow, Russia

**Keywords:** self-assembling peptides, magnetic nanoparticles, cell-penetrating peptides, triple negative breast cancer

## Abstract

Self-assembled peptides possess remarkable potential as targeted drug delivery systems and key applications dwell anti-cancer therapy. Peptides can self-assemble into nanostructures of diverse sizes and shapes in response to changing environmental conditions (pH, temperature, ionic strength). Herein, we investigated the development of self-assembled peptide-based nanofibers (NFs) with the inclusion of a cell-penetrating peptide (namely gH625) and a matrix metalloproteinase-9 (MMP-9) responsive sequence, which proved to enhance respectively the penetration and tumor-triggered cleavage to release Doxorubicin in Triple Negative Breast Cancer cells where MMP-9 levels are elevated. The NFs formulation has been optimized via critical micelle concentration measurements, fluorescence, and circular dichroism. The final nanovectors were characterized for morphology (TEM), size (hydrodynamic diameter), and surface charge (zeta potential). The Doxo loading and release kinetics were studied in situ, by optical microspectroscopy (fluorescence and surface-enhanced Raman scattering–SERS). Confocal spectral imaging of the Doxo fluorescence was used to study the TNBC models in vitro, in cells with various MMP-9 levels, the drug delivery to cells as well as the resulting cytotoxicity profiles. The results confirm that these NFs are a promising platform to develop novel nanovectors of Doxo, namely in the framework of TNBC treatment.

## 1. Introduction

Cancer is one of the most lethal diseases and tumor-related morbidity and mortality have significantly increased in recent years with breast cancer (BC) being the most frequent cancer in women worldwide. Three main biological subtypes with distinct genetic, molecular, and clinical relevance have been defined: those expressing estrogen (ER+) and progesterone receptors (PR+), those overexpressing human epidermal growth factor receptor 2 (HER2+), with or without ER+, and those defined by the absence of all these receptors, called triple-negative breast cancer (TNBC) [1,2]. ER and HER2 are also therapeutic targets, which may be exploited in addition to radio/chemotherapy if needed. TNBC has no targeted therapy as no druggable molecular targets have been identified yet and is therefore treated only with radio/chemotherapy [3]. TNBC represents between 10 and 20% of all breast cancers, has high relapse rates early after initial therapy, and has currently the worst prognosis of all subtypes [4]; thus, the new biological and target agents have to be identified starting from the basic characterization and subtypes. Therapeutic options are mainly limited to surgery, radiation therapy, and conventional chemotherapy using highly toxic compounds with a consequent high incidence of unwanted side effects on healthy tissues. The low efficiency is also responsible for de novo and/or acquired resistance to classical chemotherapeutic regimens [5]. As a matter of fact, the development of therapy-resistant metastases in vital organs, in particular the liver, lung, and brain, is the main cause of death in relapsing patients [6]. Actually, there are some new treatments in the pipelines and the first targeted therapy for metastatic TNBC has been approved in April 2020 [7]. Among the classical drugs used in TNBC treatment, Doxorubicin (Doxo) is one of the most widely used, including the liposomal form. Doxo vectorization reduces its cardiotoxicity and increases the sensitivity of cancer cells to the drug [8]. In one recent clinical trial combining Abraxane (Paclitaxel) with carboplatin or carboplatin and bevacizumab, most of the participants had serious adverse effects with no benefits [4]. The current scenario of clinical trials in TNBC is rather grim and improving treatments remains an overwhelming clinical need [9].

Nanotechnology is expected to have a significant impact, reducing the total amount of the administered drug while increasing the amount delivered to the target, thereby limiting the emergence of drug resistance and allowing the use of highly toxic compounds. Targeted nanocarriers are expected to improve their use and efficacy through multiple mechanisms: (i) protecting drugs from degradation, (ii) modifying pharmacokinetics and drug tissue distribution profiles, (iii) enhancing drug absorption, (iv) improving intracellular penetration and distribution, (v) offering novel imaging opportunities with high performances (specificity, sensitivity, contrast) for the in vivo cancer diagnosis and drug biodistribution. Novel drug delivery tools exploit a combination of chemotherapy and other moieties that are loaded in the same nanocarrier with the advantage of a timely and spatially controlled co-delivery of well-defined relative amounts of several compounds, intended to trigger synergistic effects via different pathways and should therefore increase their individual efficacy [10]. To develop precise tumor treatments, on-demand strategies exploiting the tumor microenvironment features represent an important opportunity. Nanomaterials spontaneously changing their shape and size based on specific physiological stimuli may increase drug diffusion and accumulation. These features include the overexpression of some enzymes only in certain tumors and the slightly acidic microenvironment (pH 6.0–7.2) which is related to the increased glycolysis and production of lactic acid.

Self-assembled peptide vectors customized with multiple components to optimize drug delivery efficiency represent an alternative strategy in the development of biomaterials with great potential in disease diagnosis and treatment [11]. Self-assembly in supramolecular materials is an innate ability of peptides, that helps to overcome the low peptide stability which often severely hinders their application in biomedicine. Self-assembly is a relatively straightforward strategy in which peptides spontaneously organize and convert into structurally well-defined and stabilized arrangements via non-covalent interactions due to the balance between the amphiphilic components themselves and the interaction with their environment. The peptide self-assembling process is influenced by solution conditions (such as ionic strength, pH, and assembling rate) as well as a variety of driving forces (electrostatic interactions, van der Waals interactions, hydrophobic and hydrophilic interactions, hydrogen bonds). The self-assembled peptides confer resistance to protease degradation and improve the physiological stability of therapeutics. Self-assembling sequences may be attached to other functionalities; making it relatively easy to tune the activity by simply changing the moieties on the surface without modifying the self-assembled nanostructure. The versatility of the platform paves the way to an easy modulation of properties and of the addressed pathology changing the number and/or the nature of the different moieties. We previously developed a self-assembled peptide fiber with antibiofilm activities [12], that is made of one or more structural peptides, characterized by the presence of an amino acid sequence of aliphatic residues containing a lipidic tail (C19) attached to the ε amino group of a terminal lysine to generate a peptide amphiphile. The N-terminus of the structural peptide is covalently linked to the moieties that are designed to be on the surface of the carrier, while hydrophobic moieties are encapsulated in the fiber.

In this study, the new nanofiber platform has been designed for being applied as a nanovector of anti-TNBC therapeutics, namely of doxorubicin (Doxo). The presence of the peptide gH625 as delivery moiety on the surface of the vector increases cellular uptake and endosomal escape of drugs. gH625 is a cell-penetrating peptide (CPP) able to traverse biological membranes and enhance the efficient transport of different cargoes promoting temporary lipid membrane-reorganizing processes [13,14]. gH625 has opened up an avenue in the research of peptides able to interact with membrane bilayers and to be exploited for drug delivery [15,16,17], being a promising solution to reach intracellular uptake of the nanovectors. Doxo was covalently bound to the fiber surface through an on-demand strategy which allows us to release the drug to the target site. The on-demand strategy used herein exploits the changes in the local environment of cancer tissues, which will enable active delivery of the drugs to the cancer tissues/cells to favor the drug release in the target site. In particular, we exploited the presence of over-expressed matrix metalloproteinase 9 (MMP-9) introducing an MMP-9-specific cleavage sequence between the Doxo and the fiber.

In the matrix of our fibers (hypothesized to be around 12 nm in diameter), we encapsulated superparamagnetic iron oxide nanoparticles (SPIONs) of a mean diameter of about 6 nm. The SPIONs allow us to modulate the nanovectors biodistribution (via magnetoporation) and /or drug release (via magnetic heating) using an external alternating magnetic field and to follow their accumulation in tumors with magnetic resonance imaging (MRI). MRI is one of the most powerful, non-invasive medical imaging methods, and SPIONs used as MRI imaging agents increase its sensitivity. Thus, multifunctional vectors containing SPIONs can be used in a cancer theranostic context [18].

Biodegradation will be the last step in the stimuli-responsive behavior of the nanofibers: after disassembling, their components can be rapidly cleared through the kidney [19]. Figure 1 shows the schematic composition/structure and the presumable action mechanism of the nanofibers made of self-assembled peptides, with or without SPION that have been produced in the present work.

The nanofibers were characterized in terms of size, morphology, surface charge, uptake by cancer cells, the release of Doxo in the presence of active MMP-9, and the related cytotoxicity. To our knowledge, no similar MMP-9-responsive peptide-based self-assembled carriers have been reported yet and the developed self-assembled peptide vectors may provide a method to achieve the desired personalized medicine for the treatment of several pathologies.

## 2. Materials and Methods

### 2.1. Materials

The conventional amino acids, Fmoc-Ala, Fmoc-Lys(Boc)-OH, Fmoc-Arg(Pbf)-OH, Fmoc-Gly, Fmoc-His(Trt)-OH, Fmoc-Leu, Fmoc-Ser(tBu)-OH, Fmoc-Thr(OtBu)-OH, Fmoc-Trp(Boc)-OH, Fmoc-Tyr(tBu)-OH, Fmoc-Asn(Trt)-OH, Fmoc-Ile-OH, Fmoc-Phe-OH, Fmoc-Cys(Trt)-OH, Fmoc-Pro-OH, were acquired from GL Biochem Ltd. (Shanghai, China). N,N′-diisopropylcarbodiimide (DIC), Oxyma pure, 1-[Bis(dimethylamino)methylene]-1H-1,2,3-triazolo[4,5-b]pyridinium 3-oxid hexafluorophosphate (HATU), N,N-Diisopropylethylamine, triisopropylsilane (TIS), 1,1,1,3,3,3-Hexafluoro-2-propanol (HFIP), nonadecanoic acid (C19) were purchased from Sigma-Aldrich (Italy). Rink amide p-methylbenzhydrylamine (MBHA) resin, Fmoc-L-Lys(Mtt)-OH, piperidine, and trifluoroacetic acid (TFA) were purchased from Iris Biotech GmbH (Marktredwitz, Germany). Anhydrous solvents [N,N-dimethylformamide (DMF) and dichloromethane (DCM)], Doxo-EMCH, matrix metalloproteinase-9 (MMP-9) and dialysis tubing benzoylated, Nile Red, and Thioflavin T were purchased from Sigma-Aldrich (Milan, Italy). Silver nitrate (AgNO**_3_**), and trisodium citrate were obtained from Sigma-Aldrich (Luzais, Saint-Quentin-Fallavier, France). Ultrapure water was produced using a Barnstead EASYpure RoDi system (Thermo Fisher Scientific, Villebon sur Yvette, France). All chemicals and reagents were of analytical grade and used as received.

### 2.2. Peptide Synthesis and Purification

The set of peptides (see the sequences in Table 1) were synthesized using Rink amide resin as solid support. The Fmoc protecting group was removed by the treatment with a solution of 30% piperidine in DMF (2 × 10 min). Fmoc-Lys(Mtt)-OH was used as the first amino acid for each peptide to perform the conjugation of the lipidic tail (nonadecanoic acid, C19) on the amine group in the lysine side chain. Each reaction coupling was performed through two coupling steps. In the first one, Fmoc-amino acid (4 eq) was added with N,N′-diisopropylcarbodiimide (DIC, 4 eq) oxyma pure (4 eq) as coupling reagents, in DMF for 25 min at rt; in the second one, Fmoc-amino acid (4 eq) was added with HATU (4 eq), DIPEA (8 eq), in DMF for 25 min at rt [20,21]. After the assembly of the peptide sequence, the Mtt group of the lysine was removed [22] to covalently conjugate the tail C19 on the side chain. The Mtt deprotection was performed by treating the resin with the cocktail of DCM:TFA:TIS (94:1:5, *v*/*v*/*v*), 20 × 2min at rt. Once the complete Mtt removal was ascertained by the colorimetric Kaiser test used for the detection of primary amines in a solid phase, the lipid tail C19 was performed using nondecanoic acid (2 eq), DIC (2 eq), oxyma pure (2 eq) in NMP for 2h at rt. The C19 coupling was repeated using HATU (2 eq) and DIPEA (4 eq), in NMP for 2h at rt.

The peptides were cleaved from the resin with an acid solution of TFA/H_2_O/TIS (95/2.5/2.5, *v*/*v*/*v*) and in presence of cysteine residues, 1,2-ethanedithiol (EDT) was added (TFA/water/EDT/TIS, 94/2.5/2.5/1, *v*/*v*/*v*/*v*,). After 3 h, peptides were precipitated in ice-cold diethyl ether, separated by centrifugation (2 × 15 min, 6000 rpm), and freeze-dried overnight. The peptides were dissolved in H_2_O (0.1% TFA) and HFIP (10%) and were purified by RP-HPLC (Shimadzu Preparative Liquid Chromatography LC-8A) on a Phenomenex Jupiter 4 μm Proteo 90 Å 250 × 21.20 mm column, with a linear gradient of solvent B (0.1% TFA in acetonitrile) in solvent A (0.1% TFA in water) from 10 to 90% in 25 min. The yields in purified compounds were approximately 40% for all peptides. The purity and molecular weight of the peptides were determined using LTQ-XL Thermo Scientific linear ion trap mass spectrometer. HPLC spectra and mass measurements are reported in Appendix A.

### 2.3. Synthesis and Characterization of Superparamagnetic Iron Oxide Nanoparticles

SPIONs were synthesized as aqueous ferrofluids by a coprecipitation of ferric and ferrous chlorides in an alkaline medium. Briefly, magnetite nanoparticles were precipitated by adding ammonia solution (30 mL, 35%) to an aqueous mixture of Fe^3+^ (0.032 mol FeCl_3_, 350 mL H_2_O) and Fe^2+^ (0.016 mol, FeCl_2_ 20 mL HCl 1.5 M) salts. To stabilize the chemical composition of SPIONs (magnetite/maghemite ratio), after the co-precipitation step, the SPIONs were additionally oxidized using ferric nitrate, thus increasing the surface layer of maghemite. Finally, the SPIONs were peptized in nitric acid and re-suspended in a determined volume of water. The next step consisted of coating the SPIONs with a polysiloxane layer (Sil.SPIONs). For that, 2.20 mL (12.4 mmol) of APS (3-aminopropyltrimethoxysilane) in 10 mL of methanol were added to a mixture of 20 mL (8.8 mmol of iron) of SPIONs and 10 mL of methanol. The mixture was stirred at room temperature for 12 h. To the resulting solution, 20 mL of glycerol was added and methanol then water was removed with a rotary evaporator. After evaporation, the solution was dehydrated in a vacuum at 100 °C for 2 h. The treated nanoparticles were washed three times with 40 mL of water/acetone mixture (30/70 *v*/*v*). Following the addition of 40 mL of water, peptization was performed by slowly decreasing Ph to 3 with 1 M nitric acid under vigorous stirring [23,24]. The SPIONs were characterized by dynamic light scattering techniques, while determining their average hydrodynamic diameter (DH) and surface charge (ζ-potential), using Zetasizer Nano-ZS (Malvern Instruments, Worcestershire, UK). All the measurements were performed in triplicate.

### 2.4. Conjugation of Peptide P4 with Doxo and Characterization by Nuclear Magnetic Resonance Spectroscopy NMR

The cysteine residue in the N-terminus of peptide P4 was exploited for the binding with the maleimide derivative of the Doxo (Doxo-EMCH). In particular, 600 μL of Doxo-EMCH (2.4 mM) in water/dimethylformamide (1:1 *v*/*v*) was mixed gently with 600 μL of peptide P4 (1.2 mM) in PBS using an electromagnetic agitator for 1.5 h at 27 °C in the dark. Doxo-EMCH was added dropwise at 25 μL/min [25,26]. Then, the crude compound was lyophilized and purified by dialysis with the membrane MWCO Da11000.

The reaction between P4 peptide and Doxo-EMCH was followed by NMR spectroscopy; NMR spectra were recorded on a Varian Unity Inova 600 MHz spectrometer equipped with a cold probe. Spectra were acquired at 298 K on samples at roughly 200 µM concentration and volumes equal to 500 µL in either DMSO (Dimethyl-Sulfoxide-d6, 99.9% D, Sigma-Aldrich, Milan, Italy) and in D_2_O (Deuterium Oxide 99.9% D, Sigma-Aldrich, Milan -. 2D [^1^H, ^1^H] TOCSY (Total Correlation Spectroscopy) [27] experiments were recorded with mixing times equal to 70 ms with a number of scans ranging from 16 to 64 scans, 128–256 FIDs in t1, 1024 or 2048 data points in t2. NMR spectra were processed with VNMRJ 1.1D (Varian by Agilent Technologies, Milan, Italy); 2D TOCSY spectra were analyzed with the software NEASY [28] enclosed in CARA (http://www.nmr.ch/) (accessed on 12 April 2022).

### 2.5. Formulation of Self-Assembled Nanosystem

To perform experiments, protocols for the preparation of mother solutions of peptides and of their co-assembly into nanofibers were set up. Peptide stock solutions were prepared with two different protocols which were compared. Peptides were dissolved at the highest concentration of 200 µM in water (first protocol) and in HFIP (second protocol). Different aliquots were taken to prepare aqueous solutions of the single peptides or peptide mixtures at different concentrations. In the co-assembled mixtures (P1 + P2, P1 + P3), the peptide molar ratio was 1:1, in trimeric fiber (P1 + P2 + P3) the peptide molar ratio was P1:P2:P3 = 1:0.5:0.5, while in the Doxo containing fiber the peptide molar ratio was P1:P2:P3:P5 = 1:0.8:0.05:0.15 and was kept constant for all experiments. Then, in the first protocol, the solutions were diluted until the concentration was lower than the critical concentration of aggregation and sonicated for 15 min to break any type of pre-existing aggregates; for the second protocol, the solutions were diluted with 1 mL of water and sonicated. All the samples were freeze-dried and hydrated with the proper volume of water, buffer, or dye, in order to obtain the desired concentrations. All the solutions were left to equilibrate for 1 h before use. The two preparations were compared through the calculation of critical aggregation concentrations (CAC) and TEM experiments.

To prepare the nanofibers containing SPIONs, the peptides were dissolved in water in separate solutions at the highest concentration 200µM. The mixtures were prepared to have a µM molar ratio P1:P2:P3:P5 = 1:0.8:0.05:0.15 with a total concentration of 200 μM. The solution was diluted until the concentration was lower than the CAC and sonicated for 15 min to break any type of pre-existing aggregate; then 2 μL of Silanized SPIONs (229 μM of iron) were added at the highest peptide concentration and the solution was freeze-dried.

### 2.6. Critical Aggregation Concentration (CAC) Determination

CACs of self-assembling peptides were determined by a fluorescence assay with Nile red (NR), a solvatochromic fluorescent probe. NR is poorly water-soluble while displaying a large preference to partition in aggregates that present hydrophobic binding sites and produce a blue shift and hyperchromic effect, which was measured. A methanolic NR solution was prepared at 500 nM. All the samples were freeze-dried and hydrated with the proper volume of NR solution to obtain the desired dye concentrations for the CAC determination by spectrofluorometry. Before fluorescence measurement, all the solutions were left to equilibrate for 1 h. Emission spectra for each solution were measured by a Cary Eclipse Varian spectrometer. The NR emission spectra (exc wavelength 550 nm, emission wavelength range 570 to 700 nm) were measured at least in triplicate for each solution. The data were analyzed by plotting the maximum emission fluorescence corresponding wavelength (y) as a function of peptide concentration (x) and fitting with the sigmoidal Boltzmann equation (OriginPro Program for graphs):(1)$$y=\;\frac{A1-A2}{1+e^{(x-x_{0/\;\Delta x)}}}+A2$$

In the equation, A1 and A2 are two variables corresponding to the upper and lower limits of the sigmoid, respectively. Whereas x_0_ and Δx indicate the inflection point and the steepness of the sigmoid, respectively.

### 2.7. Zeta Potential Measurements

The zeta potential of different NFs was determined by Zetasizer Nano-ZS (Malvern Instruments, Worcestershire, UK). The measurements were conducted at 25 °C, at pH 7.2. All measurements were performed in triplicate for each sample (Table 2).

### 2.8. Structural Characterization by Transmission Electron Microscopy (TEM)

Morphological characterization was performed to analyze the self-assembled peptide nanostructures. Solutions of peptides and their mixtures were freshly prepared in ultrapure water at a concentration higher than their respective CACs. Then, three microliters of each sample were placed on the formvar carbon-coated grid for 1 h and washed with distilled water (three times 10 s). For negative contrast, the samples were incubated in a 2% water solution of uranyl acetate (3 × 10 s, 10 µL) and left to dry in a small drop of the last solution. The micrographs were obtained from JEM 1011 (Jeol Ltd., Tokyo, Japan) equipped with a Gatan digital camera driven by Digital Micrograph software (Gatan Inc., Pleasanton, CA, USA) at 100 kV.

### 2.9. Self-Assembled System Aggregation Analyses by Thioflavin T Assay

Thioflavin T (ThT) is a benzothiazole dye that exhibits enhanced fluorescence (exc/em at 450/482 nm) upon binding with aggregated peptides. The ThT fluorescence intensity was measured at 25  °C, before and after its addition to nanofibers, using a Varian Cary Eclipse fluorescence spectrometer. Samples were excited at 450 nm (slit width, 5 nm) and fluorescence emission was recorded at 482 nm (slit width, 10 nm). The NFs were generated by rehydrating the lyophilized peptide samples with 300 µL of water and keeping them for 1h before the measurements. The stock solutions of ThT and of the self-assembled peptide nanofibers were prepared in PBS, at 1.5 × 10^−3^ M and used to prepare the analytical solutions by dilution in deionized water. Fluorescence signal was recorded for 25 µM ThT alone and in presence of self-assembled nanosystem (P1 + P2 + P3, P1 + P2 + P3 + P5 and P1 + P2 + P3 + P5 + SPION). Controls were recorded using the same concentration of nanosystem in PBS, without ThT.

### 2.10. Enzymatic Cleavage-Mediated Release of Doxo: Assay in Solution

Doxo release from the self-assembled fiber P1 + P2 + P3 + P5 (fiber total concentration 400 μM and Doxo concentration 100 µM) was evaluated using the matrix metalloproteinase-9 (MMP-9). In particular, the fiber was prepared as reported before and hydrated in the following buffer solution:50 mM HEPES, 200 mM NaCl, 10 mM CaCl**_2_**, and 1 mM ZnCl**_2_**, at pH 7. The MMP-9 used for cleavage was pre-activated by APMA 100 µM and Tris-HCl 50 mM (pH 7.2) and was left at 37 °C for 3 h [29].

For the release test, 50 µL of the pre-activated enzyme (40 nM) and 100 µL of the self-assembled nanosystem were mixed to prepare samples kept in a thermostatic bath at 37 °C for 30, 60, and 90 min. At the scheduled time intervals, the samples were centrifuged at 13.000 rpm for 30 min and the supernatant was analyzed by UV/vis spectroscopy (NanoDrop™ 2000/2000C, Jasco, Milan, Italy) following absorbance at 480 nm (Doxo). Doxo release was evaluated in absence of MMP-9 at pH 7, 3, and 10.

### 2.11. Circular Dichroism (CD) Analysis

Self-assembled nanosystem solutions for CD studies were prepared at different concentrations of total peptides and hydrated with water or an appropriate hydration solution for each experiment to be conducted (concentration, ionic strength, and pH). CD spectra were recorded from 195 nm to 260 nm in a Jasco J-810 spectropolarimeter using a 1.0 or 0.1 cm quartz cell at room temperature under a constant flow of nitrogen gas. Other experimental settings were as follows: scan speed of 5 nm/min, the sensitivity of 50 mdeg, time constant of 16 s, and bandwidth of 1 nm. Each spectrum was obtained by averaging three scans and converting the signal to mean molar ellipticity. CD measurements were carried out for the different peptides alone or in combinations at different concentrations, ionic strength, and pH values.

### 2.12. Preparation of Silver Plasmonic Nanoparticles for SERS Experiments

Aqueous colloids of citrate-coated AgNPs were obtained by heat-mediated reducing of silver in the presence of an excess of trisodium citrate, according to a standard protocol described by Lee and Meisel [30]. Briefly, 90 mg of silver nitrate was dissolved in 500 mL of pure water and heated until boiling. Then, 10 mL of trisodium citrate (1% m/V) was added droplet by droplet under constant agitation and the solution was kept boiling for 1 h. Silver colloid formation leads to the appearance of a characteristic green-brown color. To protect the suspension from light, aluminum paper was put around the glass vial.

### 2.13. Fluorescence Confocal Spectroscopy and Spectral Imaging (CSI)

To study the uptake of the nanofibers by cancer cells in vitro, the intrinsic fluorescence of the Doxo was followed by means of the confocal spectral imaging (CSI) technique. Briefly, the CSI consists in recording a full fluorescence spectrum from each point of a scanned optical section of the selected cell and then generating the maps (pseudo-color images) reflecting the distribution of spectral parameters (position, shape, width, intensity) in correlation with biochemical events (accumulation, interaction, metabolism, etc). The cells (SKBR3 or MDA-MB-435 cancer cell lines) were plated at a density of 5 × 10^4^ cells/well onto cover glasses in 24-well plates for 1 or 6 h. After incubation (1 to 6 h) with nanoprobes preliminary adjusted to the Doxo concentration of 5 µM, the cells were washed thrice with PBS and mounted between slide and slipcover. They were analyzed under the 50× LWD objective (numerical aperture 0.75; Olympus, Tokyo, Japan) of a LabRam laser scanning confocal microspectrometer (Horiba SA, Villeneuve d’Ascq, Hauts De France, France) equipped with a 491 nm laser source (Cobolt Calypso™), an automated X–Y–Z scanning stage, a low dispersion grating (300 grooves/mm) and an air-cooled EM CCD detector. The confocality was insured due to a pinhole of 200 µm. The laser power on the samples did not exceed 0.5 mW. No samples photodegradation was observed under the conditions used. For fluorescence spectral images acquisition from living cells, the living cells were placed under a 50× microscope and scanned through their equatorial optical section (scanning step of 0.8 µm). The full fluorescence spectrum was recorded from each scanned point (typically 30 × 30 spectra per cell, 0.05 s per spectrum). The same microspectrometer was also used in a non-confocal mode (pinhole of 1000 µm), through a 10× microscope objective (numerical aperture 0.50; LM Plan Fl, Olympus, Japan), in order to record statistically relevant spectra from solutions/suspensions.

The specific fluorescent spectrum of free Doxo at 5 µM in an aqueous solution was compared with that of self-assembled nanofibers (P1 + P2 + P3 + P5 with a concentration of Doxo of 15 µM) dispersed in water. Each spectrum was recorded as an average of 9 scans of 0.1 s. The spectra presented in the figures are the averages of at least three independent measurements. Both measurements and data treatment were performed using the LABSPEC software, version5, by Horiba Scientific (Longjumeau, France).

### 2.14. SERS Spectra Acquisition

Similar to fluorescence, SERS spectra measurements were carried out using a LabRAM confocal microspectrometer (Horiba Jobin-Yvon, Longjumeau, France) using a 690 nm diode laser source. For SERS, the Ag NPs were aggregated by the addition of 50% of PBS buffer pH 7.4 and mixed with 10% volume of the sample. SERS spectra (region from 300 to 1750 cm^−1^) were recorded using a 5 µL droplet placed under a 10× microscope objective of the microspectrometer. Spectra presented in the figures are averages of at least three independent measurements. Both experiment control and following data treatment were performed using the LABSPEC software package.

### 2.15. Cell Culture

The cytotoxicity profiles of the NF-Doxo and NF-Doxo-SPION were established on cancer cell line MDA-MB-231 (triple negative human breast cancer). Triple MDA-MB-231 (ECACC, Salisbury, UK) cells were cultured at 37 °C in an atmosphere containing 5% CO**_2_**. The culture medium was made of DMEM supplemented with 10% fetal bovine serum, 1% non-essential amino acid (Hyclone Laboratories, Logan, UT, USA), and 1% penicillin/streptomycin (Gibco**^®^**, Life Technologies, Paisley, UK). The cell harvesting was made with trypsin/EDTA (0.05%) (Gibco**^®^**, Life Technologies, Paisley, UK) at 80% of confluence.

### 2.16. Cell Proliferation Assays

Cell viability and proliferation were studied, using a luminescent test based on quantification of ATP, using the CellTiter-Glo cell proliferation assay (Promega, Madison, WI, USA). Briefly, 3.000 MDA-MB-231 cells were incubated in 100 μL of medium in 96-well plates for 24 h and then treated with different samples. A H_2_O_2_ solution at 10 mM was used as positive control and the culture medium alone was tested as a negative control. Doxo (stock solution at 5.24 mM in PBS) was used as a reference. Cells were treated with NF-Doxo or NF-Doxo-SPION diluted in culture media from 0.05 to 5 µM in Doxo. Cells were incubated with 100 μL of each solution at 37 °C with 5% CO_2_ for 72 h. Cell viability was then determined using CellTiter-Glo reagent (Promega, Madison, WI, USA). Briefly, 100 µL of medium were removed and 100 μL of CellTiter-Glo reagent were added to each well. The plates were shaken for 2 min and then incubated at room temperature for 10 min. The luminescence values were measured with a gain at 135, with an acquisition at 0.5 s, using a microplate reader (Bio-Tek**^®^** instruments, Inc., VT, Santa Clara, CA, USA). When a dose-dependent activity was observed, 50% inhibitory concentration (IC50) was calculated using Graphpad PRISM 7 software (San Diego, CA, USA) (*n* = 4 in quadruplicate).

### 2.17. Effect of the Magnetic Field on the Nanofibers Containing SPIONs

The effect of the magnetic field was studied on SPION-free and SPION-containing nanofibers, without or with 5 µM P3. A range of 4 µL drops of the 4 samples was placed on a glass slide parallel to the rectangular-shape stationary magnet and allowed to dry overnight at room temperature. In this way, the magnetically-attracted SPIONs and nanofibers would form a deposit with a gradient increasing on the magnet side. The aggregated material deposit observable with white light illumination under a 10 objective of the LabRaman confocal microspectrometer was interpreted as particle- and/or fiber-related. The Doxo fluorescence observable with 491 nm laser irradiation was interpreted as nanofiber-related. For each drop, two rectangular zones of the same dimension (35 × 230 µm), one situated on the side close to the magnet and another on the side opposite to the magnet, were scanned to record a spectral map of Doxo fluorescence (0.1 s per spectrum, 5 µm step between spectra). The maps were used to calculate the average fluorescence intensity of the zone.

## 3. Results

### 3.1. Peptide Design and Synthesis

Here, we report a strategy for the design of nanofibers (NFs) made of self-assembling amphiphilic peptides, P1, P2, P3, and P5 (Table 1 and Figure 2).

### 3.2. Peptide Screening by MIC for Candida Species

As shown in Figure 2, the peptide amphiphiles were constituted of: (i) a biologically active hydrophilic sequence covalently linked to a hydrophobic moiety, composed of an alkyl chain (C19, made of 19 carbon atoms) and (ii) a hydrophobic amino acid sequence. The alkyl chain and the peptide sequence were carefully chosen to optimize the properties of the formed nanofibers. Once in the polar aqueous medium, these hydrophobic moieties were intended to self-assemble forming the nanofiber core. The multi-alanine sequence increases the hydrophobic driving force and further favors self-assembly. Fine adjustment of the alanine number can lead to the formation of completely different self-assembled morphologies such as short nanorods, nanotubes, nanofibers, nanovesicles, and so on. We previously proved that the hexa-alanine sequence was likely leading to nanofiber formation [12]. To improve the nanofiber solubility in water and make it biologically active, a domain carrying positive or negative charges and a bioactive domain were incorporated.

Herein, we used 4 peptides (P1, P2, P3, and P5), each playing a definite role in the formulation and/or in biological activity. All the peptides had the same hydrophobic part (C19–lysine–hexa-alanine) but differ in the domains at the N terminal side. The P1 peptide had a negatively charged domain, while P2, P3, and P5 carried a positively charged domain at the same position. Thus, in addition to hydrophobicity, the electrostatic interactions contributed to the nanofiber formulation. The amount of positively charged P2 peptide was used to adjust the ratio of positive/negative charges, and thus stabilize the nanofiber. The other two positively charged peptides P3 and P5 provided respectively biocompatibility/bioactivity. The P3 peptide was covalently linked to the cell-penetrating peptide (CPP) gH625 which is known to favor the uptake of the nanofiber into cells. The P5 peptide carried the anticancer drug doxorubicine (Doxo) attached via a linker sequence (PLGSYL) which can be cleaved by MMP-9 enzyme over-expressed in cancer cells. Thus, the MMP-9 enzymatic activity was intended to trigger the drug release in cancer cells. The P4 peptide which has the linker, but yet does not have the drug, was synthesized as the precursor of P5. All the compounds were synthesized by solid phase peptide synthesis strategy, purified by HPLC, and then characterized by electrospray mass spectrometry (ESI-MS, see Appendix A).

### 3.3. Conjugation of Doxo-EMCH to Peptide P4 to Produce the P5 Compound

The binding of the Doxo was performed through a maleimide moiety and the side chain of a cysteine residue (Figure 3A). The reaction between the peptide P4 and Doxo-EMCH was followed by NMR spectroscopy. Comparison of 1D [^1^H] NMR spectra recorded in DMSO-d6 for Doxo-EMCH and the reaction product (P5) indicates, as expected, the disappearance of the peak arising from maleimide double bond at ~7 ppm [25,31] (Figure 3B,D) in the spectrum of P5 and appearance of signals characteristics of peptide backbone HN protons between 7.6 and 8.2 ppm along with aromatic protons deriving from the tyrosine residue present in the P4 peptide (6.7 and 7.1 ppm) (Figure 3C,D). Further investigation by 2D [^1^H, ^1^H] TOCSY spectra acquired in DMSO clearly highlights the dual nature of P5, which contains peak characteristics of P4 and in detail, the aromatic Hδ and Hε protons of the tyrosine residue, which correlate between each other, several cross peaks in between backbone amide and side chains protons of different amino acids, along also with aromatic protons from the doxorubicin system (Figure 3D). 2D [^1^H, ^1^H] TOCSY spectra were also recorded in D_2_O to reduce spectral complexity through H/D exchange (Figure 3E). In fact, the TOCSY spectrum of P5 in D_2_O is lacking signals from HN protons, that exchange with deuterium while, containing peaks from aromatic protons (Figure 3E). In major detail, the presence in the spectrum of P5 in D_2_O of characteristic aromatic protons of Doxo (around 7.5 ppm) and aromatic protons from the tyrosine in P4 peptide, and the absence of the maleimide double bond peak (at 6.6 ppm) indicate that the chemical reaction in between the P4 peptide and Doxo-EMCH led to the desired product P5. In D_2_O, very broad peaks from the aromatic system of Doxo (Figure 3E) are likely to reflect aggregation phenomena and/or interconversion between different conformers.

### 3.4. Nanofiber Formulation and Characterization

Several nanofibers were developed and characterized. Two different protocols were set up for nanofiber formulation as reported in the experimental section with no significant difference as evidenced by circular dichroism (CD) and fluorescence experiments (data not shown). The nanofibers (NFs) without external functions were obtained by mixing P1 + P2 at a molar ratio of 1:1. The nanofibers decorated with CPP were generated by mixing respectively P1 + P2 + P3 (molar ratio 1:0.5:0.5) or P1 + P3 (molar ratio 1:1). Thus, the P1 + P3 contained higher CPP molar fraction, two-fold that of the P1 + P2 + P3. To formulate the nanofibers decorated with both low CPP fraction and peptides P4/P5 bearing the linker without or with Doxo, respectively, we used respectively the mixtures P1 + P2 + P3 + P4 and P1 + P2 + P3 + P5 (NF-Doxo) at molar ratio 1:0.8:0.05:0.15 (Figure 2).

We determined the CAC (critical aggregation concentration) for the peptides, alone or co-assembled NFs while keeping an equal ratio of positive/negative charges, using a fluorescence assay with the solvatochromic fluorophore Nile red (NR). Poorly water-soluble, NR has a large preference to partition in aggregates that offer hydrophobic binding sites. The decreased polarity of the hydrophobic environment produces a hyper-hypsochromic effect (blue-shift and intensity increase) on the NR fluorescence. The data were analyzed by plotting the emission fluorescence maximum wavelength (y) as a function of peptide concentration (x) and fitting it with the sigmoidal Boltzmann equation. The plots obtained and the respective CAC values are shown in Figure 4A–F and Table 2.

The data obtained are indicative of a good tendency to self-assemble for each peptide and of an enhanced ability to co-assemble for peptide combinations in agreement with our design. The results obtained are very promising as the three peptides can co-assemble and the concentration at which the peptides are aggregated generally decreases when peptides are co-assembled, compared to the peptides alone. In particular, P1 + P2 showed a lower CAC, while P1 + P3 presented a higher CAC indicating a lower tendency to form fibers in presence of gH625 on the surface. Thus, we prepared the sample P1 + P2 + P3, which contained a lower quantity of gH625. The reduced content of gH625 favors the formation of the fiber as evidenced by the lower CAC.

The TEM experiments performed on P1 + P3 (data not shown) and P1 + P2 + P3, clearly suggest that the two-fold higher CPP content favors inter-fiber interactions which render the NFs larger, apparently too much large for their eventual applications as injectable vectors. Instead, the data obtained on P1 + P2 + P3 showed the formation of fibers with a diameter of ca. 12 ± 2 nm and a length of ca. 150 ± 50 nm (Figure 4I). These dimensions seem appropriate for applications in drug delivery according to the extravasation criterium. Indeed, it is known that for extravasation from leaky blood vessels of tumors, the nanoobjects need to have at least one dimension below the size of fenestrations (200–800 nm, depending on the tumor site and type) present in the vessel walls. We thus decided to continue the study with a lower CPP molecular fraction in the NF formulation. The ThT assay on P1 + P2 + P3 showed a significant increase in the fluorescence intensity (Figure 4H), indicating peptide aggregation and thus further supporting the formation of the NF.

Once characterized the vector assembly, we also analyzed the properties of the fibers conjugated to Doxo (NF-Doxo, made of P1 + P2 + P3 + P5) or not (made of P1 + P2 + P3 + P4). The data relative to the CAC determination and CD spectroscopy (Table 2 and Figure 5A,B,D) suggests that the presence of Doxo decreases the concentration at which the peptides with Doxo are aggregated while it does not affect the secondary structure of the peptides in the final NF. Furthermore, the ThT assay on NF-Doxo (Figure 5E) also indicated aggregation supporting the formation of the fiber. The TEM data (NF-Doxo, Figure 5F) confirmed the formation of fibers.

To better unravel the role of gH625 on the size and morphology of nanofibers, we compared (Figure 6A,B) the TEM images of the formulation (NF-Doxo) with and without gH625 (P1 + P2 + P5; in this formulation without gH625, P3 is substituted by the same concentration of P2 to keep the correct ratio between positive and negative charges and the same concentration of Doxo on the fiber surface). While both the NF morphology and diameter remained nearly constant (12 ± 2 nm), in the absence of gH625, it was possible to observe more elongated fibers of 200 ± 50 nm instead of 150 ± 50 nm. The presence of gH625 seems also to favor the formation of more defined and smaller fibers which should be further adapted for biomedical applications.

The effect of the CPP on the NF length was even more significantly affected once we included in the formulation protocols SPIONs (equivalent of 229 μM of iron concentration). SPIONs were first prepared according to a protocol previously defined as reported in the experimental section. We used different protocols and decided to add the SPIONs to the peptide amphiphiles prior to the lyophilization and hydration step, with the aim of obtaining more stable colloidal nanosystems. In particular, the obtained mixtures were evaporated to complete dryness and then re-hydrated with water.

We compared the zeta potential of the NF-Doxo, and NF-Doxo-SPION was similar, supporting the hypothesis that the SPIONs are located on the surface or just below the surface of the fiber and thus are not modifying the charge of the fiber. Furthermore, TEM data (Figure 6C,D) clearly showed that SPIONs are located on the surface or just below the surface of the fiber, which confirmed our design strategy and opens the way to the possible application of SPIONs to direct the nanosystem. Interestingly, panel C reports fibers with Doxo and SPION and without gH625 (P1 + P2 + P5 + SPION), while panel D reports fibers with Doxo and SPION and with 5 μM gH625 (P1 + P2 + P3 + P5 + SPION, NF-Doxo-SPION). Both images show that the aggregation of fibers is lower in presence of SPIONs and confirms the effect of gH625 on the morphology of the fibers which become shorter (50–150 nm instead of 200–600 nm) and thicker (15 ± 2 nm instead of 12 ± 2 nm) than in absence of gH625.

### 3.5. NF-Doxo Fibers Stability

Effects of NF-Doxo concentration and the aqueous environment parameters (ionic strength and pH) on their structural stability were analyzed by CD spectroscopy. For that, peptide mixtures at a total peptide concentration of 400 μM were prepared in water, lyophilized then re-hydrated and diluted in a series of solutions. The results of CD spectroscopy indicated that the nanofibers were preserved upon the dilution: the spectra were enough overlapped even near the CAC value (Figure 7A).

Furthermore, to probe the effect of ionic strength (variable in body fluids), the NF-Doxo samples at a total peptide concentration of 200 μM were re-hydrated with NaCl solutions (salt concentrations from 0 to 5 mM). The CD spectra shape did not change significantly (Figure 7B), thus indicating good stability of the system.

The effect of pH on the stability of NF-Doxo was analyzed at pH 3, 7, and 10. We observed very similar CD spectra for the fibers at pH 7 and 10 (Figure 7C), while at pH 3 their slight perturbation might be in favor of an eventually accelerated fiber dissociationupon pH decrease, a condition typical of cancer tissues.

### 3.6. Drug Release in Aqueous Solution

Doxo release from the NF-Doxo was evaluated using UV-Vis spectroscopy. Before each measurement, the solution was centrifugated to eliminate the nanofiber from the sample. The experiment was performed in triplicate. The enzyme-free or the enzyme-mediated Doxo release was assessed at 37 °C, in the absence or in presence of 40 nM of MMP-9, a metalloproteinase usually over-expressed in breast cancer cells. The results obtained at pH 7 show a relatively rapid release of around 76 ± 1% at 30 min followed by a sustained and slow release over a prolonged period of time with a release of 82 ± 1% and 84 ± 2% at 60 and 90 min respectively (Table 3). While the Doxo release was accelerated in presence of MMP-9 at pH 7, at enzyme-free conditions no release takes place at pH 7, 3, and 10.

### 3.7. Comparison of Doxo Spectra in NFs vs. Free Drug Solution

Fluorescence. Due to its chromophore made of conjugated aromatic rings (π ≥ π* transitions) substituted with two carbonyl and two phenol groups, the Doxo molecule possesses an intrinsic fluorescence emission. This emission is known to be solvatochromic. Figure 8A compares the fluorescence spectra of Doxo, the free drug in an aqueous solution, and the drug covalently attached to the surface of the NFs (NF-Doxo). The Doxo fluorescence further supports the previously reported NMR data confirming the covalent bond formation between the Doxo and linker molecules (Figure 8A). The fact that the fluorescence spectra are rather similar in shape is concomitant with the design strategy that in the NF-Doxo is not encapsulated within the fiber but is rather exposed to the aqueous media on the fiber surface. However, some minor changes of the spectral shape (an increase of the shoulder intensity ratio at 639/560 nm) observed on going to the NFs might indicate a partial change of its environment and/or effect of the covalent binding. To go further in studying the drug situation on the NFs, we analyzed the NF aqueous suspensions by SERS spectroscopy.

SERS. Surface-enhanced Raman scattering (SERS) is a potent optical spectroscopy technique [32], combining high sensitivity close to that of fluorescence and high molecular specificity of Raman. Indeed, the SERS spectrum, made of multiple narrow bands, represents a unique vibrational signature of molecules and enables their selective recognition [33]. To obtain the SERS response, the molecules (analytes or Raman reporters) have to be adsorbed on a plasmonic surface like that of silver nanoparticles: the resonance of incident light with the surface plasmons of a noble metal [34] leads to the enhanced Raman response. Being a short-range effect, the SERS signal can be obtained only for the chromophores accessible to the plasmonic NP surface. In the present study, the characteristic SERS signal of Doxo recorded from the NF-Doxo suspension (Figure 8B) confirmed that the drug was on the NF surface and not buried inside of the fibers, again confirming our design. Some minor changes in the SERS spectrum of Doxo once it is bound to fibers are also confirming the data of fluorescence and NMR.

### 3.8. NF-Doxo Uptake by Cancer Cells

We analyzed the uptake of the NFs by following the Doxo fluorescence in different compartments of living cancer cells. To this scope, we performed confocal spectral imaging (CSI) spectroscopy. Unlike conventional laser-scanning confocal fluorescence microscopy, CSI consists in recording the total emission spectrum from each point scanned. Much more relevant than the overall intensity measurement at a given wavelength range, the fluorescence spectral shape analysis in CSI enables us to map the specific molecular events underwent by the fluorescent drug, like its interaction with intracellular compartments or environment changes following the release from a vector.

No spectra characteristic to free Doxo in aqueous solution or to initial NFs in aqueous suspension were found in the cells (Figure 9). On the contrary, we observed spectra typical of Doxo complexed with either nuclear DNA or with cytosolic membranes/proteins. In the case of the nanofibers carrying the drug (NF-Doxo), that result is indicative of drug release from the NFs once the latter is uptaken into the cells.

In particular, the Doxo molecule is known to efficiently reach the nucleus of the treated cells, the nuclear DNA binding of Doxo by intercalation being the main mechanism of anticancer action of this drug. Due to the drug intercalation between the DNA base pairs (π orbitals interaction), the nuclear Doxo fluorescence shows the characteristic bathochromic shift of ca. 12 nm, with the main emission maximum going from ca 597 to 609 nm (see the red spectrum in Figure 9). In contrast, in cytosolic regions of a low-polarity environment, like that of cellular membranes and/or hydrophobic domains of proteins, Doxo fluorescence undergoes rather a hypsochromic shift, with an increase of the intensity ratio of the shoulders at 560/640 nm (green spectrum in Figure 9). Therefore, we used the fluorescence spectra of these two molecular complexes of Doxo as model spectra. There were also some points of cytosol in which the environment was less hydrophobic, but those points represented a very weak fraction of the total fluorescence and thus we decided to neglect them for simplifying the description. Each intracellular spectrum was deconvoluted in a proportional sum of these two spectra. The deconvolution coefficients were then used to generate the corresponding maps of the drug interaction in cells.

We comparatively studied the internalization of free Doxo and that delivered by NFs, either without or decorated with 1 µM and 5 µM P3 (gH625) on their surface. The data obtained for the MDA-MB-231 cell line incubated for 1 h are reported in Figure 9. As shown, nuclear fluorescence (red spectra, red zones in images, and red histograms) was dominant for the cells treated with free Doxo solution and those treated with NF decorated with P3 (gH625), without significant difference between 1 and 5 µM of the CPP. In contrast, in the cells treated with NF-Doxo void of gH625 (P1 + P2 + P5), the drug showed the dominant fluorescence intensity from the hydrophobic cytosolic regions. Thus, decoration of the NF with P3 enhanced the nuclear delivery of Doxo and presumably favors the anticancer effect of the drug. In addition, the higher concentration in CPP led to enhanced uptake of the drug, as shown by the average fluorescence intensity which increases by nearly a factor of 3 on going from NFs decorated with 1 µM to 5 µM of gH625.

### 3.9. Cell Proliferation Assays

The cell viability profiles shown in Figure 10 and the IC50 values shown in Table 4 were obtained on the tumor cell line MDA-MB-231 (triple negative human breast cancer), incubated with free Doxo solution at 5 µM, NF with (NF-Doxo-SPION) and without SPION (NF-Doxo) loaded with 5 µM of Doxo and with NF (without SPION and DOXO). The IC50 values for cells incubated with free Doxo, and with NF-Doxo demonstrate that the nanofiber is capable of inducing cell death in tumor cells, with IC50 cell viability 10 times lower for the cells treated with free Doxo. This can be explained by the delay of Doxo release from the fibers compared to Doxo in the solution.

### 3.10. Effect of the Magnetic Field on Nanofibers Containing SPIONs

The effect of the magnetic field was studied on SPION-free (NF-Doxo) and SPION-containing nanofibers (NF-Doxo-SPION), without or with 5 µM P3. For this, we simply observed the process of air-drying of 5 µL droplets in presence of the stationary magnet. If there is an effect of the magnetic field on the samples, we would observe a deposit with a gradient where magnetically-attracted SPIONs and/or nanofibers will be more densely present on the magnet side. The images in Figure 10 show the two sides of each deposit, the one opposite to the magnet (Figure 10B) and the one close to the magnet (Figure 10C). As can be seen from the white-light images of the zones, for the samples containing SPIONs and not those free of SPIONs, there was a higher presence of the nanofiber on the side closer to the magnet. Clearly, this is indicative of SPIONs migrating to the magnet upon drying. Interestingly, the SPION-containing deposits showed aggregates that more or less followed the magnetic field lines. To confirm that not only the SPIONs but also the NFs were moving as a consequence of the magnetic field, we followed the change in Doxo fluorescence in the deposits. An increase in Doxo fluorescence towards the magnet was also evidenced for the SPION-containing samples. While for the nanofibers without SPIONs, independently form the presence or absence of gH625 (P3), the fluorescence intensity was nearly the same (±17%) on both sides of the droplets, it increased by +221.6% and +651.2%, respectively, on the side close to the magnet for both the NF-Doxo-SPION without and with gH625 (P3). This result suggests that the magnetic field acted not only on the SPIONs but was also able to attract a fraction of the nanofibers attached to SPIONs.

## 4. Discussion

The concept of NFs for drug delivery is explored in our laboratory using peptide amphiphile molecules, which can self-assemble into cylindrical nanofibers easily decorated on their surface with the desired moieties for achieving the desired bio-application (for a comprehensive review on the use and advantages of peptides for anticancer therapies see ref [35]). Here, it was possible to develop NFs with Doxo and gH625, both loaded from 1 to 15 µM per 100 µM of total peptide concentration. The Doxo was covalently linked to the fiber via a stimuli-responsive linker which was released in an MMP9-dependant manner. Furthermore, the introduction of SPIONs to the formulations enabled the samples to be sensitive to the magnetic field.

The amount of the four peptides in the NF allowed us to modulate the NF properties like size and colloidal stability as well as the consequent ability to enter the cancer cells and deliver Doxo to the nucleus. Clearly, the amount of bioactive peptide used is limited to allow for unaffected assembly of the structure. In our case, we observed a clear dependence of the aggregation, morphology, and colloidal stability on the amount of gH625 present on the fiber surface. The α-helical gH625, when present at high concentrations, favors inter-fiber interactions which reduce the colloidal stability of the formulation. We thus, explored the minimum quantity of gH625 necessary on the fiber surface to enhance the internalization of the vector. We found that even at low concentrations between 1 and 5 μM the NFs internalization is significantly enhanced. Another key issue for internalization is the shape of the nanosystems and their dimensions. Elongated and flexible biomaterials such as nanofibers might have a therapeutic advantage over traditional spherical nanoparticles owing to their prolonged circulation time in vivo [36]. For instance, Ben Akiva et al. [37] fabricate anisotropic PLGA nanoparticles which presented a similar fluidity or stability to spherical nanoparticles and could be covered by naturally derived cell membranes, and were able to better escape macrophage clearance, showing a superior half-life compared to any other spherical nanoparticles. Indeed, it is known that for extravasation from leaky blood vessels of tumors, the nanoobjects need to have at least one dimension below the size of fenestrations (200–800 nm, depending on the tumor site and type) present in the vessel walls. The TEM results clearly indicated that the lower content of gH625 favors the formation of more defined and smaller fibers which should be more adaptable for biomedical applications. Interestingly the TEM also showed that SPIONs were located on the surface or just below the surface of the fiber, which confirmed our design strategy and opens the way to the possible application of SPIONs to direct the nanosystem. Furthermore, the aggregation of fibers is lower in presence of SPIONs and confirms the effect of gH625 on the morphology of the fibers which consequently become shorter and thicker. The number of SPIONs incorporated was clearly sufficient to observe a deposit with a gradient increasing on the magnet side and clearly evidenced that the magnetic field was acting not only on the SPIONs but also on the NFs. Our nanofiber with a diameter of ca. 12 ± 2 nm and length of ca. 150 ± 50 nm was tested on TNBC cells as a model, and we calculated an IC_50_ of 1.215 µM that is 10 times lower than Doxo (0.1542 µM).

## 5. Conclusions

In this study, a proof of concept is provided showing how nanoscale engineering can be exploited to produce peptide-based supramolecular platforms with anticancer activity for tumor personalized therapy. The use of naturally inspired peptide self-assembled materials, owing to their excellent biodegradability, bioactivity, environmental sensitivity, and active sites for chemical modification, exhibit several advantages for drug delivery, such as temporally and/or spatially controllable drug release.

In conclusion, we demonstrated that an accurate and efficient peptide design in the framework of the rational formulation strategy is a potent tool to provide new peptide-drug delivery systems which can be further functionalized with organic molecules and/or inorganic imaging agents like SPIONs, to produce a new generation of targeted theranostic nanovectors. These potentialities of the NFs are currently being further explored in our laboratories.

## Figures and Tables

**Figure 1 pharmaceutics-14-01544-f001:**
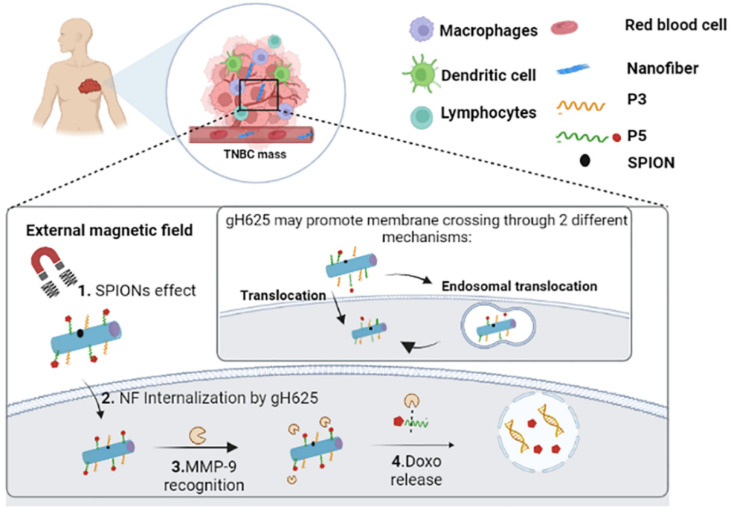
Schematic mechanism of action of the nanofibers. This Figure was created with BioRender.com (accessed on 28 December 2021).

**Figure 2 pharmaceutics-14-01544-f002:**
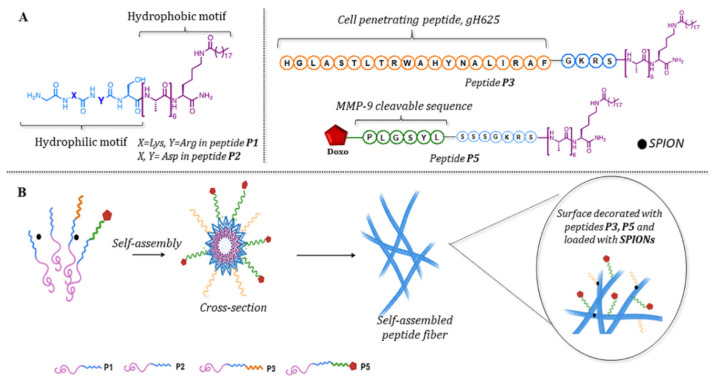
Schematic representation of peptides involved in nanofiber composition (**A**) and the potential self-assembly process leading to the organization in nanofibers (**B**). This figure was created with BioRender.com (accessed on 1 January 2022).

**Figure 3 pharmaceutics-14-01544-f003:**
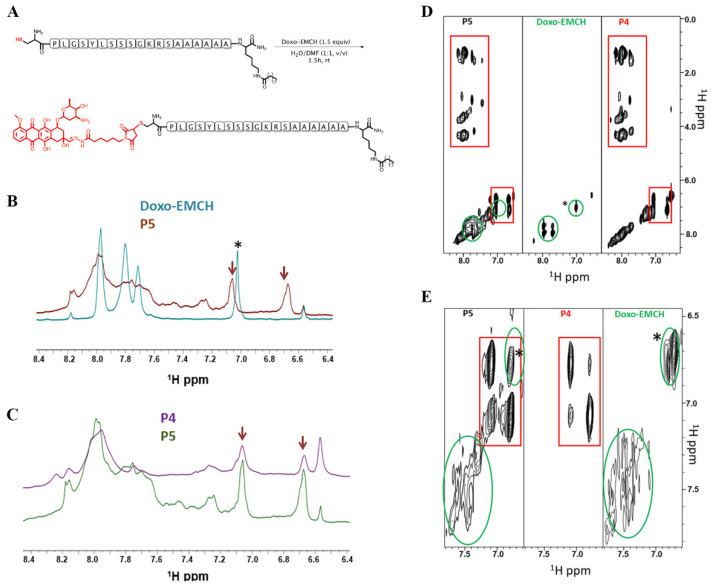
Synthetic strategy for achieving the conjugation of peptide P4 to Doxo (**A**). Comparison between 1D [^1^H] NMR spectra recorded in DMSO of Doxo-EMCH (light blue) and P5 (Red). The * and arrows are pointing to the double bond in the maleimide ring and the tyrosine aromatic protons in the P4 peptide, respectively (**B**). Comparison of 1D [^1^H] NMR spectra acquired in DMSO of P5 (green) and P4 (violet) peptides, signals from aromatic protons of tyrosine are indicated by arrows. Chemical shifts were referenced to the DMSO signal at 2.55 ppm (**C**). Comparison of 2D [^1^H, ^1^H] TOCSY spectra of P5 (left side, **D**), Doxo-EMCH (central, **D**) and P4 peptide (right side, **D**). The spectral region containing correlations including HN and aromatic protons are shown. The red rectangles highlight signals from the P4 peptide whereas, green cycles indicate peaks from Doxo-EMCH. The * points to a peak deriving from protons of the maleimide double bond (**D**). 2D [^1^H, ^1^H] TOCSY spectra recorded in D_2_O of P5 (left side, **E**), P4 (central, **E**), and Doxo-EMCH (right side, **E**). Red rectangles indicate peaks from aromatic protons in the tyrosine of the P4 peptide. Green cycles encompass peaks from Doxo-Maleimide. Chemical shifts were referenced with respect to residual water peak at 4.75 ppm (**E**).

**Figure 4 pharmaceutics-14-01544-f004:**
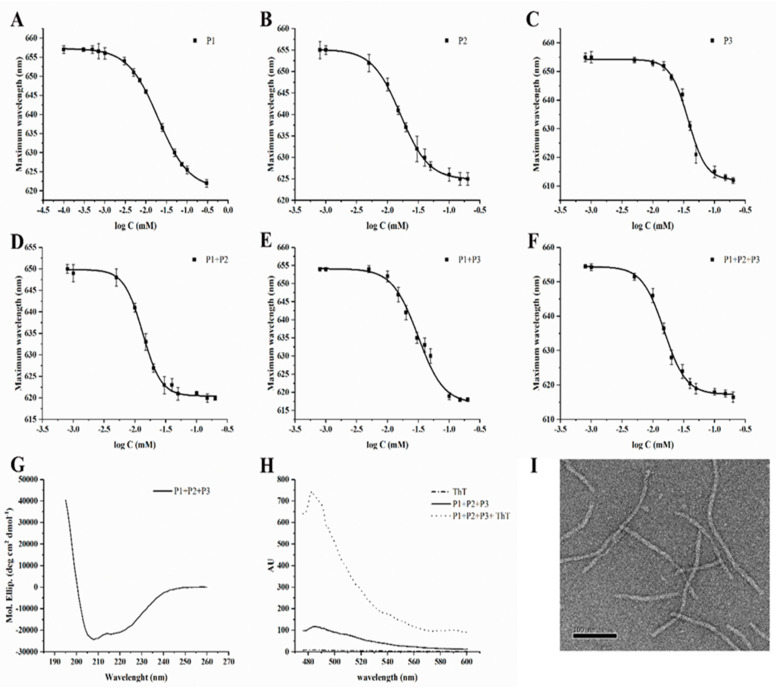
Fiber formation results: (**A**–**F**) report the critical aggregation concentration for P1, P2, and P3 alone, in their binary and ternary complexes; (**G**) reports the CD spectrum of P1 + P2 + P3 mixture; (**H**) reports the ThT fluorescence spectra increase in presence of aggregates of the P1 + P2 + P3 mixture; (**I**) shows the typical morphology of the P1 + P2 + P3 mixture as seen in TEM with negative contrast. Graphs were created with OriginPro 2021.

**Figure 5 pharmaceutics-14-01544-f005:**
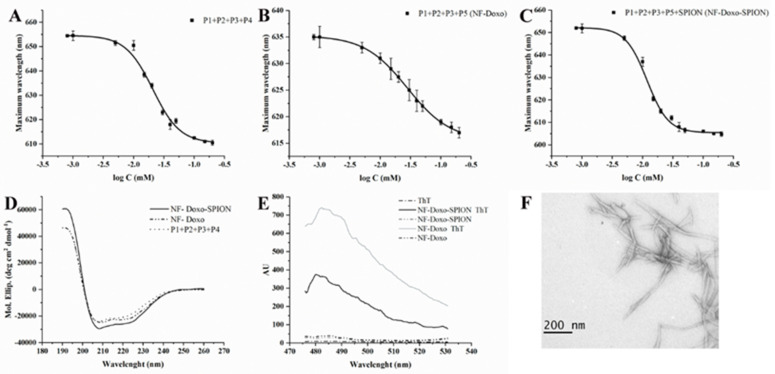
Fiber formation and characterization: CAC (**A**–**C**), CD results (**D**), Tht results, (**E**) TEM (**F**) for NF-Doxo, and NF-Doxo-SPION. Graphs were created with OriginPro 2021.

**Figure 6 pharmaceutics-14-01544-f006:**
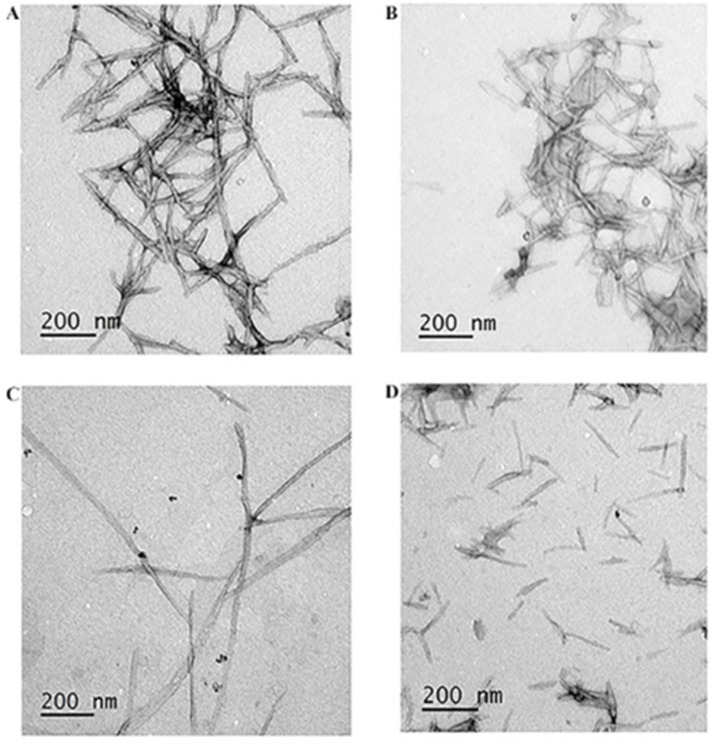
TEM images for P1 + P2 + P5 (**A**), NF-Doxo (P1 + P2 + P3 + P5) (**B**), P1 + P2+ P5 + SPION (**C**), NF-Doxo-SPION (P1 + P2 + P3 + P5 + SPION) (**D**).

**Figure 7 pharmaceutics-14-01544-f007:**
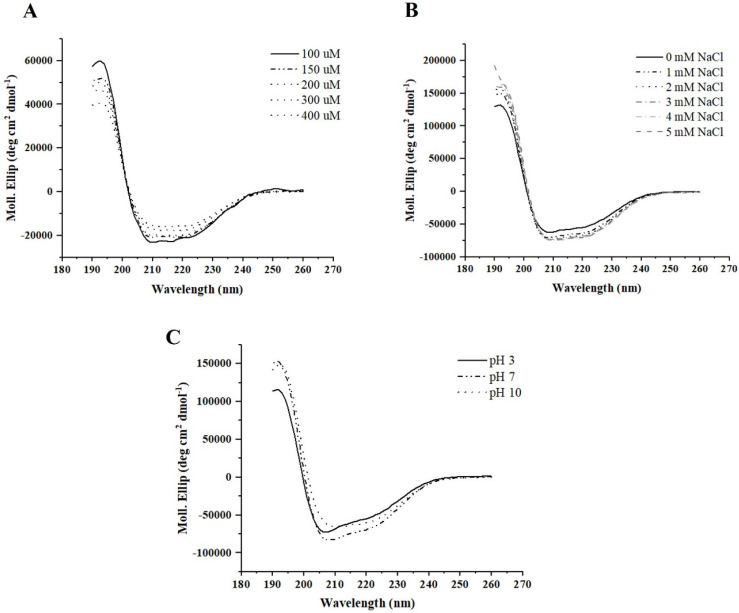
CD spectra of nanofibers NF-Doxo: effects of dilution (**A**), ionic strength (**B**), and pH (**C**). Graphs were created with OriginPro 2021.

**Figure 8 pharmaceutics-14-01544-f008:**
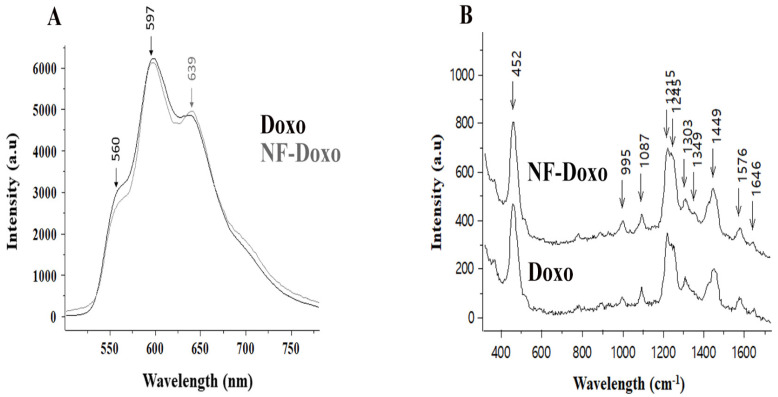
Fluorescence (**A**) and SERS (**B**) spectra of Doxo: free drug in aqueous solution compared with NF-Doxo in aqueous suspension.

**Figure 9 pharmaceutics-14-01544-f009:**
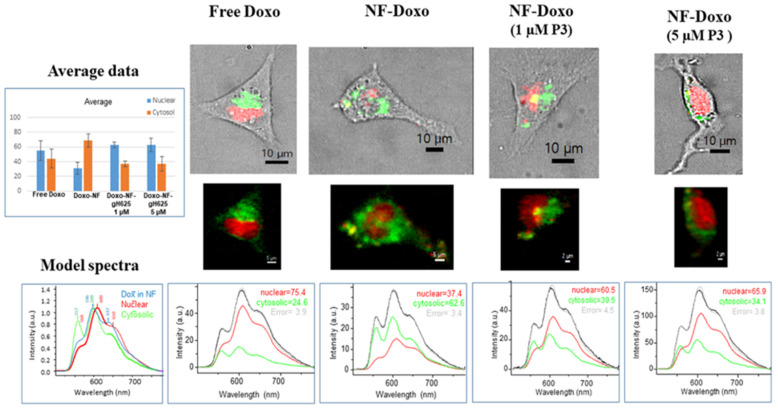
Results of fluorescence confocal spectral imaging on MDA-MB-231 cells treated for 1 h with 5 M solution of free Doxo in solution and with **NF-Doxo**, decorated with 1 or 5 µM of **P3** or void of **P3**. Left top window: summarized statistical data on the two fractions of intracellular fluorescence: nuclear (red) and cytosolic (green). Top panels: typical spectral maps and their overlay with video images of the cells. Bottom panels: Window on the left side: model spectra of fluorescence–Doxo in **NF-Doxo** (blue); Doxo in nuclear zones (red); Doxo in cytosolic zones (green). Middle and right-side windows: average spectra of the cell shown above deconvoluted into the two model spectra (cytosolic and nuclear).

**Figure 10 pharmaceutics-14-01544-f010:**
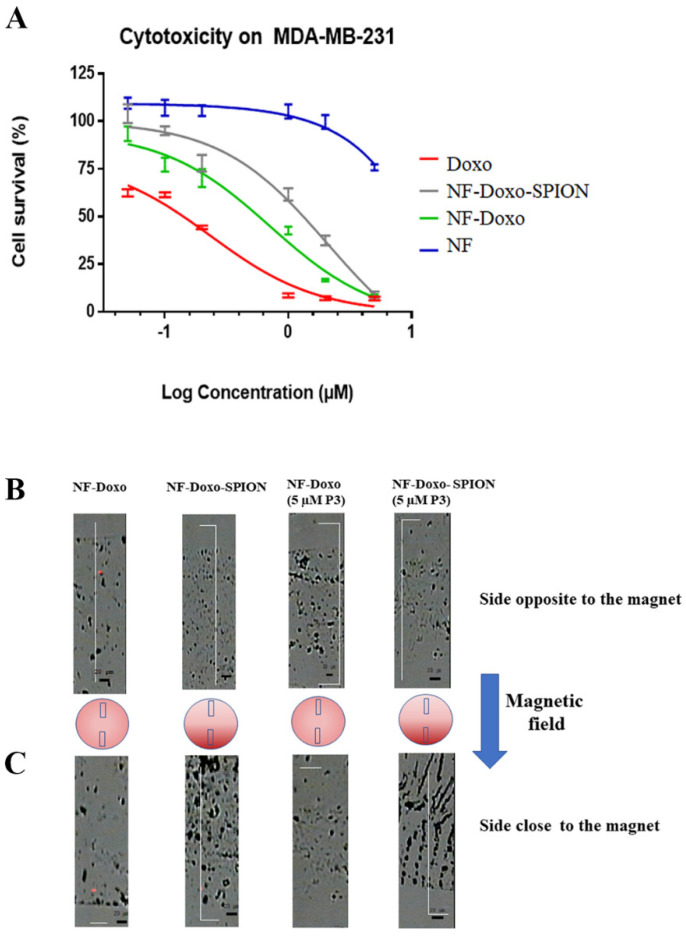
Comparison of cytotoxicity plots on breast cancer cells of the nanofibers (**A**). Images recorded by LabRaman confocal microspectrometer on two sides of the drop analyzed for each sample, on the side opposite to magnet (**B**) and another on the side close to the magnet (**C**).

**Table 1 pharmaceutics-14-01544-t001:** Sequences of peptides P1–P5 forming the nanofiber.

Peptide	Sequence
P1	NH_2_-GDDS-AAAAAA-K(C19)-CONH_2_
P2	NH_2_-GKRS-AAAAAA-K(C19)-CONH_2_
P3	NH_2_-HGLASTLTRWAHYNALIRAF-GKRS-AAAAAA-K(C19)-CONH_2_
P4	NH_2_-CPLGSYL--SSS-GKRS-AAAAAA-K(C19)-CONH_2_
P5	NH_2_-C(Doxo)-PLGSYL-SSS-GKRS-AAAAAA-K(C19)-CONH_2_

**Table 2 pharmaceutics-14-01544-t002:** Formulations and Critical Aggregation Concentrations (CAC).

Formulation	Peptide Ratio	CAC (μM)	Zeta Potential (mV)
P1	-	20.1 ± 1.0	ND
P2	-	16.3 ± 1.0	ND
P3	-	37.3 ± 1.0	ND
P1 + P3	1:1	31.3 ± 1.0	ND
P1 + P2	1:1	13.3 ± 0.9	ND
P1 + P2 + P3	1:0.5:0.5	15.2 ± 0.9	+20.7 ± 1.6
P1 + P2 + P3 + P4	1:0.8:0.05:0.15	20.8 ± 1.0	+2.55 ± 1.80
P1 + P2 + P3 + P5 (NF-Doxo)	1:0.8:0.05:0.15	28.2 ± 0.9	+10.5 ± 1.6
P1 + P2 + P5	1:0.85:0.15	76.2 ± 1.2	ND
P1 + P2 + P3 + P5 + SPION (NF-Doxo-SPION)	1:0.8:0.05:0.15	11.8 ± 0.8	+9.42 ± 1.01
P1 + P2 + P5 + SPION	1:0.85:0.15	40.9 ± 0.8	+7.11 ± 0.38
SPION	-		−21.3 ± 1.7
ND = Not determined			

**Table 3 pharmaceutics-14-01544-t003:** The percentage of Doxo released at different time points.

Time (min)	Doxo Released (%)
**30**	76 ± 1%
**60**	82 ± 1%
**90**	84 ± 1%

**Table 4 pharmaceutics-14-01544-t004:** IC_50_ (µM) values on MDA-MB-231.

Nanosystem	IC_50_
**NF-Doxo**	1.215
**NF-Doxo-SPION**	2.078
**NF**	>1000
**Doxo**	0.1542

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
