# Peer review of "Design and Validation of Nanofibers Made of Self-Assembled Peptides to Become Multifunctional Stimuli-Sensitive Nanovectors of Anticancer Drug Doxorubicin"

_pharmaceutics, 2022, doi:10.3390/pharmaceutics14081544_

Round 1

Reviewer 1 Report

The manuscript titled ‘Design and validation of nanofibers made of self-assembled peptides to become multifunctional stimuli-sensitive nanovectors of anticancer drug doxorubicin in Triple Negative Breast Cancer’ describe the development of self-assembled peptide-based nanofibers (NFs) with the inclusion of a cell penetrating peptide (namely gH625) and a matrix metalloproteinase-9 (MMP-9) responsive sequence, and its cytotoxicity profiles on TNBC models. The work is interesting and innovative. However, there are still some issues to be solved before publication.

1. The title is to treat the TNBC models. So the author should give more details for the characterizations and treatment for TNBC in the introduction part. And as we know, MMP9 is existing in more than one of the malignant tumor, not specially existing in TNBC cells. So the author should elucidate why the NFs-DOXO could specific act on TNBC cells.

2. The author used peptide amphiphile molecules, which can self-assemble into cylindrical nanofibers easily decorated on their surface with the desired moieties for achieving the desired bio-application. Why the nanorods, nanotubes or nanovesicles could not achieve such a desired bio-application. The author should explain it.

3. There are 12 figures now in the manuscript. That’s too much. The author should combine some figures into one figure. For example, the author could divide the figures into four parts: synthesis, characterization, in vitro assay and in vivo assay, that will be more clear and concise.

4. The author should design the assay to prove the responsive ability of MMP9 and the targeting ability of the CPP should be quantified. It is very important for NF’s characterization.

5. Please supplement the missing figure in the results of drug release.

6. Please supplement the experiment to prove the beneficial effect of SPIONs on TNBC model treated by NFs.

7. We need authors to provide experimental data related to cytotoxicity to verify whether the nanofiber platform has good biosafety.

8. Please make your conclusions more concise and show only the high impact outcomes. Report your conclusions in one or maximum 2 paragraphs.

9. Some format mistakes:

① Check all formats,e.g. CaCl2,ZnCl2 in Page 7 Line 303,2 should be in subscripted,e.g. 1.5×10-3 in Page 7 Line 294,-3 should be in superscripted,please check all carefully.

② Please check the description of matrix metalloproteinase-9 and use MMP9 or MMP-9 uniformly.

③ Check the use of spaces and multipliers,(e.g. Page 4 Line 167;e.g. Page 6 Line 251),please check all carefully.

④ Please supplement the missing data in Table 2 or give the explanation.

Reviewer 2 Report

Dear Authors,

The reviewed work entitled: Design and validation of nanofibers made of self-assembled peptides to become multifunctional stimuli-sensitive nanovec-tors of anticancer drug doxorubicin in Triple Negative Breast Cancer by Del Genio et. al. concerns the interesting topic of obtaining and characterizing peptide-base nanofibers with SPIONs nanoparticles to better visualistion (MRI) and biodistribution (via magnetoporation).

Methodologically, the work is very well prepared; the synthesis carried out, the purification and also the study of the properties of the proposed combination are correct. Admittedly, the procedure of peptide synthesis is quite intricate and can probably be shown schematically, but this is an additional addition rather than a substantive objection. My curiosity is aroused by the efficiency of the reaction of synthesis of these peptides, was it determined and if so at what level? The method of producing conjugates from Doxo and SPION is also not objectionable. The techniques for characterizing the various components of the developed nanosystems are well chosen. It is unfortunate that in subsection 2.5 Formulation of self-assembled nanosystem there is no diagram showing the combinations of peptides and active nanosheets used to prepare a particular nanosystem. This would have made it easier to distinguish and follow the nanosystems during their characterization. Figure 2 is not clear for me

What are the parameters A1, A2 and how does A2 differ from A2 in the equation used to determine CAC? How do the magnetic properties of SPIONs manage to be maintained in fibers if they are subjected so many times to chemical procedures of combining them with peptides and purification or if they are in aqueous solutions; it is known that iron oxides quickly oxidize losing their magnetic properties, are they somehow protected during chemical treatment?

The drawing is not clear to me as to the approach in combining/combining the synthesized peptides (peptide fibers), throughout the work I was wondering which materials are actually being studied; whether only pure synthesized peptide-base nanofibers (and if so, of what composition) or any of the peptides in combination with SPION and/or Doxo. A little bit of light is shed by Table 2, but in the rest of the paper it is not quite clear whether always the same set of materials is tested or only selected ones .A good scheme would be a significant help here.

In general, the work in my opinion is interesting, however, the basic and main complaint is the lack - absolute lack - of discussion of the results in relation to the literature, to works on similar topics using this technique of forming fibers and combining them with nanoparticles or anticancer drugs. Scientific papers aiming to show an innovative approach must be based on and relate to what has already been done/researched showing that the new approach has further greater/better capabilities. Here it is difficult to say whether the results obtained are comparable, better or worse. This is a methodically well-prepared research report, but without a reference to the literature data can not be published as a paper. after this supplementation, the work is most suitable for publication

Round 2

Reviewer 1 Report

I think the current version can be received directly.